# Classification of Breast Lesions on DCE-MRI Data Using a Fine-Tuned MobileNet

**DOI:** 10.3390/diagnostics13061067

**Published:** 2023-03-11

**Authors:** Long Wang, Ming Zhang, Guangyuan He, Dong Shen, Mingzhu Meng

**Affiliations:** Department of Radiology, The Affiliated Changzhou No.2 People’s Hospital of Nanjing Medical University, Changzhou 213164, China

**Keywords:** mobile convolutional neural networks, deep learning, breast lesions, magnetic resonance imaging

## Abstract

It is crucial to diagnose breast cancer early and accurately to optimize treatment. Presently, most deep learning models used for breast cancer detection cannot be used on mobile phones or low-power devices. This study intended to evaluate the capabilities of MobileNetV1 and MobileNetV2 and their fine-tuned models to differentiate malignant lesions from benign lesions in breast dynamic contrast-enhanced magnetic resonance images (DCE-MRI).

## 1. Introduction

In 2020, female breast cancer surpassed lung cancer as the most commonly diagnosed cancer [1,2]. It is also one of the leading causes of death for women [3,4]. Breast cancer must be detected early and correctly diagnosed as malignant or benign to prevent its further progression and complications. By doing so, it is possible to plan an effective and timely treatment, which in turn lowers the mortality rate associated with this disease.

The traditional methods for detecting and diagnosing breast cancer have several important limitations [5,6]:

1. Experts are unavailable in remote areas (underdeveloped countries).

2. There is a lack of domain experts that are capable of accurately analysing multiclass images.

3. The process of reviewing large numbers of medical images every day can be exhausting and tedious.

4. As a result of the subtle nature of breast tumours and the complexity of the breast tissue, a manual examination is challenging.

5. The concentration levels requested to medical experts and other types of fatigue make diagnosis more difficult and time-consuming.

The presence of such facts lengthens the diagnosis process and may lead to diagnostic errors. Utilizing additional methods to increase diagnosis efficiency and reduce the false prediction rate is always necessary. In recent years, artificial intelligence (AI) technology has made great progress in automatically analysing medical images for anomaly detection. In comparison with manual inspection, automated image analysis using AI reduces the time and effort needed for manual image screening and more efficiently captures valuable and relevant information from massive image collections [6,7,8,9].

Recent advances in AI research have focused on the automation of the early detection and diagnosis process in light of its importance. In addition, the advent of robust AI algorithms (deep learning methods) has led to a surge in research activities in this field. It is important to have hardware that can train/run these robust and complex AI algorithms, as well as sufficiently large datasets, for training AI algorithms. There are several imaging modalities that researchers have utilized to automate breast cancer detection, including mammography, ultrasound, magnetic resonance and histopathological imaging or any combination of these methods [3,10,11,12,13,14].

Deep learning techniques have achieved state-of-the-art results in many computer vision tasks, and the medical field is no exception [15,16,17]. Presently, most of the deep learning models used in breast cancer detection are classic techniques; their excessive number of parameters, whose data occupy considerable space, prevents their use on mobile phones or low-power devices; therefore, their deployment on low-cost mobile computers is still challenging.

In general, the deep learning models can be divided into two types according to the architecture of the convolutional neural network (CNN): lightweight models and complex models. MobileNet is a lightweight model designed specifically for mobile and embedded terminals; compared with other convolutional neural networks, it greatly reduces the required number of parameters [18].

There are several works using the CNN for other tasks. Taresh et al. evaluated the effectiveness of state-of-the-art pretrained CNN models for the automatic diagnosis of the novel coronavirus disease 2019 (COVID-19) from chest X-rays and found that MobileNet obtained the highest accuracy of 98.28% [19]. In this article, we evaluated the capabilities of MobileNetV1 and MobileNetV2 and their fine-tuned models to differentiate malignant legions from benign lesions in breast DCE-MRI images.

## 2. Materials and Methods

### 2.1. Dataset

A total of 310 patients with complete breast DCE-MRI and pathological data were collected from January 2017 to December 2020, including 17 patients with bilateral lesions (benign and malignant lesions on one side). All lesions were pathologically confirmed (via surgical or needle biopsy) in this step. Lesions were divided into benign and malignant groups. The ages, pathological types, and tumour diameters of the two groups were compared (Table 1). The inclusion criteria were as follows: I. patients who did not receive any preoperative chemotherapy or chemoradiotherapy before magnetic resonance imaging (MRI); II. no puncture or surgical procedure was performed before the MRI examination. We discarded images in which both benign and malignant lesions appeared.

To prevent the influence of image laterality on the evaluation of benign and malignant lesions on the bilateral breasts of the same patient, the examined breast DCE-MRI images were unilateral images. To eliminate the interference signal of some tissues (such as the aorta, etc.), the images were cropped (using photoshop), and the part containing the breast tissue (efforts to contain the axillary tissue) was kept. Finally, 2124 benign lesions’ images (benign group) and 2226 malignant lesions’ images (malignant group) were obtained. The images of the benign and malignant lesions groups were randomly divided into a train set (benign lesion group: 1704 images, malignant lesion group: 1786 images), a test set (benign lesion group: 210 images, malignant lesion group: 220 images) and a validation set (benign lesion group: 210 images, malignant lesion group: 220 images) according to a ratio of 8:1:1.

### 2.2. MRI Techniques

MRI was performed using two 3T MRI scanners with a dedicated breast coil in the prone position. Gd-DTPA (0.1 mmol/kg, 2.50 mL/s) was administered via elbow vein injection. A total of six phase images were acquired (one pre-contrast phase image and five post-contrast dynamic-enhancement phase images). The detailed scanning parameters are listed in Table 2. MRI was performed preoperatively and before therapy initiation.

### 2.3. Proposed Model

The computer environment was configured with Windows 10 (the enterprise version of the 64-bit operating system), an Intel (R) Core (TM) i7-10700F CPU, an NVIDIA RTX 2060 GPU and 6 GB of RAM. Other programs were closed when the model was running. To facilitate a network performance comparison, the same training and validation sets were selected for each network. The thresholds of the models were set at 0.5; if a result was ≥0.5, the lesion in the image was judged as malignant; otherwise, it was predicted as benign. The architecture of the proposed DTL model for breast lesion classification is shown in Figure 1.

### 2.4. Data Augmentation

The images were randomly shuffled using a set of programs (random function). Data augmentation was performed before model training. The original images were augmented by flipping and rotating them, so that the augmented images maintained the original medical characteristics. The utilized parameters and their values are listed in Table 3.

### 2.5. Network Structure of MobileNetV1

The main principle of a MobileNetV1 model is the application of depthwise separable convolutions, which are each made of a depthwise convolution and a pointwise convolution [20]. The kernel of the depthwise convolutional layer slides to convolve with only one input channel. The pointwise convolutional layer has a convolution kernel size of 1 × 1. The number of output matrix channels is equal to the number of convolutional kernels, while the number of input matrix channels is equal to the number of convolutional kernel channels.

### 2.6. Network Structure of MobileNetV2

In MobileNetV2, convolutional layers, bottleneck layers and an average pooling layer form the basic network structure. The structure of the bottleneck layers can be found in reference [18]; they usually include pointwise convolutions and depthwise convolutions. When the stride is 1, the input is added to the output. Another structural feature of MobileNetV2 is inverted residuals [18]. In addition, Relu6 serves as an activation function in the inverted residuals, and this function is defined according to the following expression:(1)y=ReLU(6)=min(max(x,0)6)

### 2.7. Fine-Tuning Strategies

In this study, we designed two fine-tuning strategies for MobileNetV1 and MobileNetV2: S0 and S1. In S0, all parameters were nontrainable (False) except for the parameters in the fine-tuned fully connected layers, while in S1, all trainable (True) parameters were activated, and all parameters in the fine-tuned fully connected layers participated in the model training process (Figure 2). In this way, four models were generated for our study: MobileNetV1_False(V1_False), MobileNetV1_True(V1_True), MobileNetV2_False(V2_False) and MobileNetV2_True(V2_True).

The network structure was not changed throughout the training process. We selected the parameter convergence and capacity for generalization as the primary outcome measures for the DTL models.

### 2.8. Hyperparameter Settings

We used binary cross-entropy as our loss function. To explore suitable hyperparameter combinations for DTL workflows, we trained a DTL model for each classification task and each hyperparameter combination. (Figure 3). The input image size was 224 × 224. The training process of our model for breast DCE-MRI images required 60 epochs with a batch size of 64 images. In addition, the activation functions were ReLU and sigmoid functions in the fine-tuned fully connected layers, as shown in Equations (1) and (2):(2)$$\mathrm{ReLU}\left(x\right)=f\left(x\right)=\{\begin{array}{cc} \max\left(0,x\right), & \;\;x\geq 0\\ 0, & \;\;x<0 \end{array}$$
(3)$$\mathrm{sigmoid}\left(x\right)=f\left(x\right)=\frac{1}{1+e^{-x}}$$

### 2.9. Evaluation Metrics

To compare the performance of the DTL models, the following five performance indices were calculated as the metrics in this study: accuracy (Ac), precision (Pr), recall rate (Rc), F1 score (f1) and area under the receiver operating characteristic (ROC) curve (AUC) [21]. The positive and negative cases were assigned to the malignant and benign groups, respectively. Hence, true positives (TP) and true negatives (TN) represent the numbers of correctly diagnosed malignant and benign lesions, respectively. FP and FN indicate the numbers of incorrectly diagnosed malignant and benign lesions, respectively. TP were considered true-positive samples. These were also positive samples. The mathematical formulations of Ac, Pr, Rc and F1 are as follows:(4)$$\mathrm{Ac}=\frac{\mathrm{TP}+\mathrm{TN}}{\mathrm{TP}+\mathrm{TN}+\mathrm{FP}+\mathrm{FN}}$$
(5)$$\mathrm{Pr}=\frac{\mathrm{TP}}{\mathrm{TP}+\mathrm{FP}}$$
(6)$$\mathrm{Rc}=\frac{\mathrm{TP}}{\mathrm{TP}+\mathrm{FN}}$$
(7)$$f1=\frac{2\times \mathrm{Pr}\times \mathrm{Rc}}{\mathrm{Pr}+\mathrm{Rc}}$$

The distribution of the data was not considered in Ac. F1 is a balanced metric determined by precision and recall; it is useful when there are imbalanced classes in the given dataset.

The false-positive rate (FPR) was calculated by dividing the total number of negatives by the fraction of negatives that were incorrectly classified as positive by the model. It can be evaluated as follows:(8)$$\mathrm{FPR}=\frac{\mathrm{FP}}{\mathrm{FP}+\mathrm{TN}}$$

The false-negative rate (FNR) is the fraction of positives misclassified by the model as negative divided by the total number of positives. It can be evaluated as follows:(9)$$\mathrm{FNR}=\frac{\mathrm{FN}}{\mathrm{FN}+\mathrm{TP}}$$

## 3. Results

### 3.1. Intergroup Age and Lesion Diameter Comparisons

Comparisons between the two groups revealed significant age and lesion diameter differences (*p* = 0.029 and ˂0.001, respectively).

### 3.2. Learning Curves

We found that the combination of average pooling with all other hyperparameters (Adam, learning rate = 0.001 and dropout) performed best. By analysing the metrics obtained by the models (V1_False, V1_True, V2_False, V2_True) on the dataset, we found that the accuracy of all models reached 1.00 on the train set, and the highest accuracy (V1_True) was 0.9815 on the test set, which was higher than those of V1_False (0.9749), V2_False (0.9672) and V2_True (0.9699).

Although the network architectures of MobileNetV1 and V2 are highly similar, the highest test accuracy of V1 was higher than that of V2 (Figure 4 and Figure 5). However, they demonstrated the same disadvantages, as the loss value exhibited a rising trend with the increase in the number of epochs, which revealed that they did not converge on the dataset and that the overfitting problem occurred. However, the loss value in the test set was the lowest for V1_True, which revealed that the V1_True model had a better generalization ability than the other models. The visualization of the training process (heatmaps) is shown in Figure 6.

### 3.3. Training Time and Model Size

The total number of parameters of MobileNetV1 was greater than that of MobileNetV2, while the time required for training the MobileNetV2 model was greater than that for MobileNetV1. However, the size of the file saved by MobileNetV2 was smaller than that of the file saved by MobileNetV1. For detailed data, see Table 4.

### 3.4. Cross-Validation

Fivefold cross-validation was employed to examine the model performance. The fivefold cross-validation method and results are illustrated in Figure 7 and Table 5, respectively.

### 3.5. Classification Report

The overall Pr, Rc, f1 and AUC of V1_True on the validation set were 0.79, 0.73, 0.74 and 0.74, respectively. They were higher than those of V1_False, V2_False and V2_True. Detailed information is provided in Table 6 and Figure 8.

### 3.6. Visualization of Confusion Matrices

To intuitively show the superiority of the proposed DTL model, the confusion matrices of the DTL models are presented in Figure 9. The FPR and FNR values of V1_True were 9.52% and 43.18%, respectively. The FPR and FNR values of V1_False, V2_False and V2_True were 12.38%, 13.81%, 11.90% and 11.00% and 41.36%, 55.91% and 53.18%, respectively. V1_True provided a lower FPR than the other models. The FNR of V1_True was slightly higher than that of V1_False but lower than those of V2_False and V2_True.

## 4. Discussion

Breast cancer is now the most commonly diagnosed cancer among women, with an estimated 2.3 million new cases each year, and it is the fifth most common cause of cancer-related death worldwide. Among women, breast cancer accounts for one in four cancer cases and one in six cancer deaths, ranking first for incidence in the vast majority of countries. The death rate for female breast cancer is considerably higher in developing countries than in developed countries [1]. Therefore, it is important to search for a method that can rapidly and efficiently diagnose breast lesions early.

In contrast to normal MRI, DCE-MRI provides more detailed views of soft breast tissues, so the affected breast areas can be easily identified. Literature reports have concluded that diagnosis is the most commonly performed medical task, and deep learning techniques are most often used to perform classification, but the majority of the selected studies used mammograms and ultrasound rather than magnetic resonance images as imaging modalities [3,4,22,23].

Due to the memory and computation demands of deep neural networks, for instance, ResNet, VGG and DenseNet, they are difficult to apply to embedded systems with limited hardware resources. Models such as those listed above have many parameters and require considerable space. That is, the networks of these models almost always have deep structures, which are adequate for extracting complex image features and should be compressed and accelerated. By applying depthwise separable convolutions, MobileNetV1 is a lightweight model that can decrease the number of parameters and computational complexity with a low classification precision loss [24]. In this way, the overall computation time can be significantly reduced, and this approach has a relatively high diagnostic accuracy for the medical image classification task.

Arora et al. analysed two publicly available datasets (COVID-CT scan and SARS-CoV-2 CT-Scan) retrospectively and found that the MobileNet model provided precision values of 94.12% and 100%, respectively [25]. A separate study aimed at evaluating the effectiveness of state-of-the-art pretrained convolutional neural networks for the automatic diagnosis of COVID-19 from chest X-rays (CXRs). The outcome of their study showed that MobileNet obtained the highest accuracy of 98.28% [19].

Encouraged by these results, we evaluated the capabilities of MobileNetV1 and MobileNetV2 and their fine-tuned models to differentiate malignant lesions from benign lesions in breast DCE-MRI. We found that the MobileNetV1 models had better generalization abilities than the MobileNetV2 models, and the V1_True model obtained the highest accuracy on the test set. However, the weakness of V1_True was that its Pr and AUC only reached 0.79 and 0.74, respectively, illustrating that the robustness and generalizability of the proposed model need to be further increased in future studies. In other words, two important questions we now have to answer are: can the performance of the proposed model be further improved, and how can we improve it? An additional deficiency of this study was that an overfitting problem was observed in the V1_True model. The loss value had a rising trend with the increase in the number of epochs, which revealed that it did not converge on the dataset and that the overfitting problem occurred. We will consider two main directions for solving this problem: expanding the training datasets and optimizing the model. More importantly, we observed that the proposed model only took approximately 20 min to train on the data, and the size of the saved model was only 19.4 MB. This suggests that the proposed model was easy to deploy in the mobile terminal.

Several limitations of this study need to be acknowledged. First, the number of images in the train set was relatively small, especially regarding the lack of some rare lesions. Our dataset used for training may not represent the entire population with breast disease, which may have an impact on the accuracy of the DTL model. Therefore, further analysis with larger datasets is necessary to fully test the robustness of the DTL model. Second, during routine diagnostic procedures, clinical evaluation, breast ultrasound and mammography are performed in addition to magnetic resonance imaging; however, only DCE-MRI images were examined in our study. Third, the high performance achieved by our proposed model was based on the premise of high-quality DCE-MRI images. In clinical practice, poor-quality images from other hospitals might decrease the performance of the DTL model. Therefore, high-quality DCE-MRI images obtained via the standard procedure are highly warranted. A future study might require a multicentre collaboration to obtain a sufficiently large series of data for training and testing the proposed neural network.

## 5. Conclusions

Using deep learning techniques, we propose an intelligent lightweight-assisted model to differentiate benign and malignant lesions, called MobileNet, based on our datasets with data augmentation. The results of four models were compared, and the V1_True model achieved the best performance. Low-configuration computers are compatible with this model, which is particularly advantageous for some units in which computer availability is limited. A suitable recommendation and timely referral can also be provided, yielding a high level of diagnosis for imaging and resulting in a good social benefit. Additionally, in the future, this lightweight model can be embedded in mobile devices due to its small size and few parameters, so that mobile users can themselves perform self-screening. The network should serve our lives; so, a lightweight network is very important. However, medico-legal problems, such as misdiagnosis, arisen from the use of AI in the area of medicine cannot be ignored [26]. Research project risk management rules were proposed to use these solutions. In the authors’ opinion, however, the chances of having to use them are low.

## Figures and Tables

**Figure 1 diagnostics-13-01067-f001:**
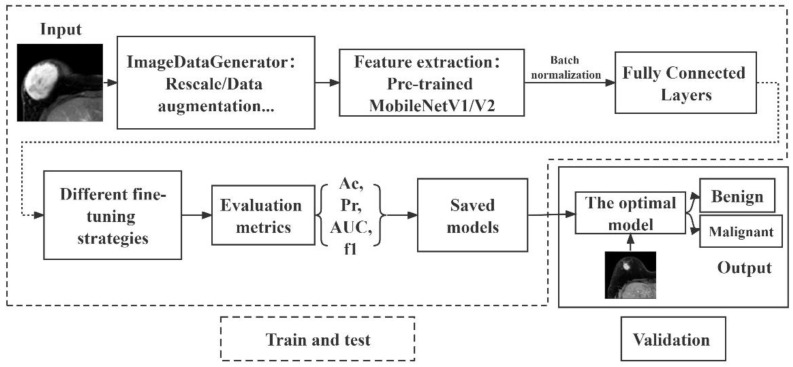
DTL diagram. The output of the DTL model is the likelihood of malignancy. The data analysis process is divided into three parts: the first part is image network feature extraction, the second part includes data training and testing, and the third part is the validation of the DTL model.

**Figure 2 diagnostics-13-01067-f002:**
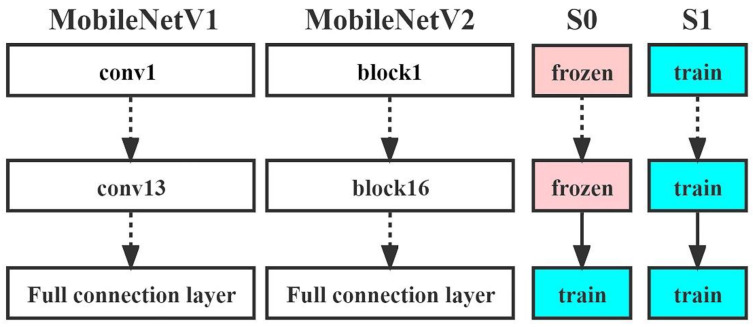
Schematic diagrams of the fine-tuning strategies for MobileNetV1 and MobileNetV2. Abbreviations used in the figure: S = strategy. Train: activated layers of the neural network; frozen: nontrainable layers of the neural network; conv: convolutional layer.

**Figure 3 diagnostics-13-01067-f003:**
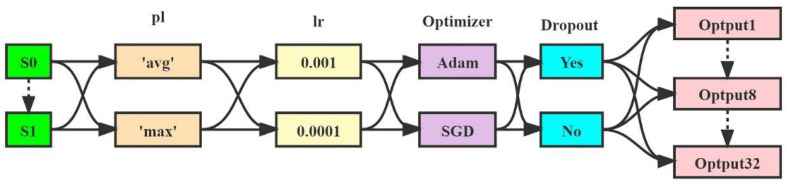
Hyperparameter settings. Abbreviations used in the figure: avg: average, max: maximum, pl: pooling, lr: learning rate, Adam: adaptive moment estimation, SGD: stochastic gradient descent.

**Figure 4 diagnostics-13-01067-f004:**
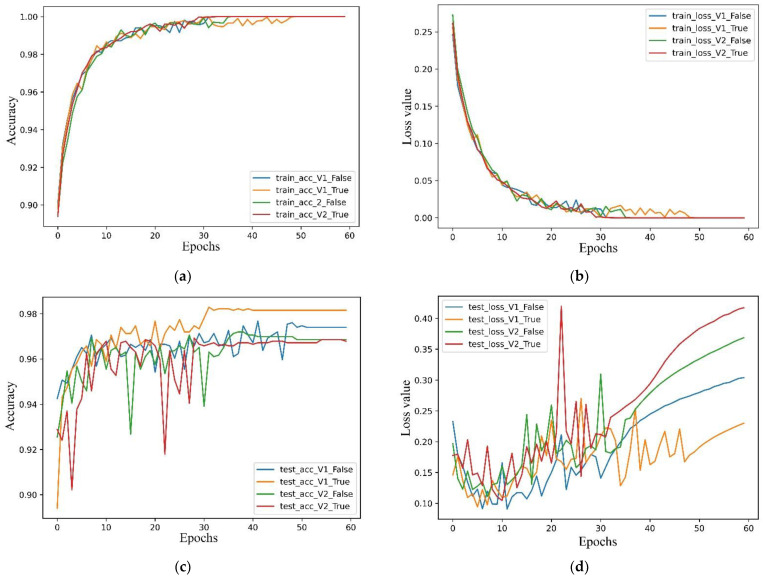
Learning curves for all models with respect to the number of epochs. (**a**) training accuracy of the proposed models, (**b**) test accuracy of the proposed models, (**c**) training loss of the proposed models, (**d**) test loss of the proposed models. The results clearly show that the V1_False model attained the highest accuracy (**b**) and lowest loss value (**d**) on the test set.

**Figure 5 diagnostics-13-01067-f005:**
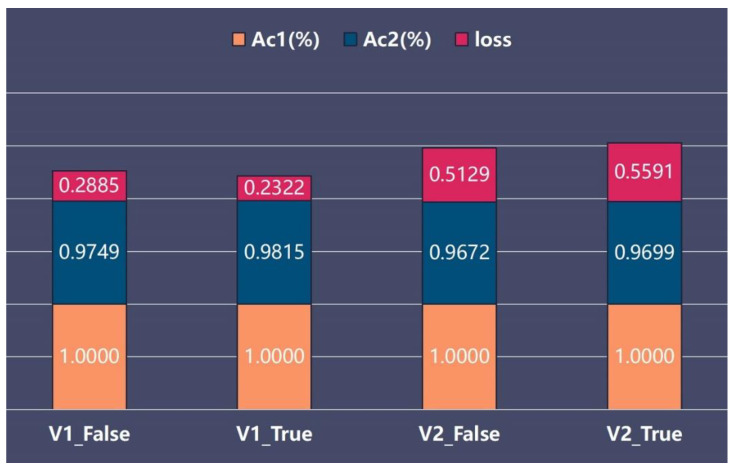
Results of model training. Abbreviations used in the figure: Ac1: accuracy on the train set, Ac2: accuracy on the test set, loss: loss on the test set.

**Figure 6 diagnostics-13-01067-f006:**
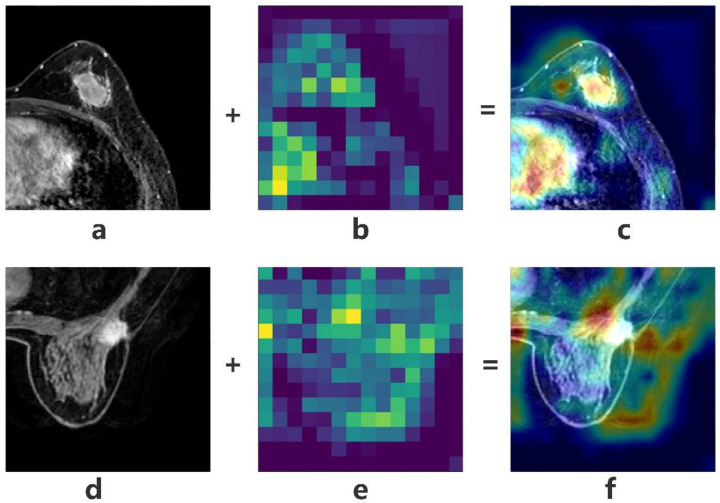
Heatmaps of V1_True. Heatmaps of activated zone boundaries in benign lesions (**a**–**c**) and malignant lesions (**d**–**f**). The heatmap of the activated zone for the malignant lesions showed greater activation than that of the benign lesions. (**a**,**d**) original images, (**b**,**e**) heatmaps of (**a**,**d**), respectively, (**c**) fusion image of (**a**,**b**), (**f**) fusion image of (**d**,**e**).

**Figure 7 diagnostics-13-01067-f007:**
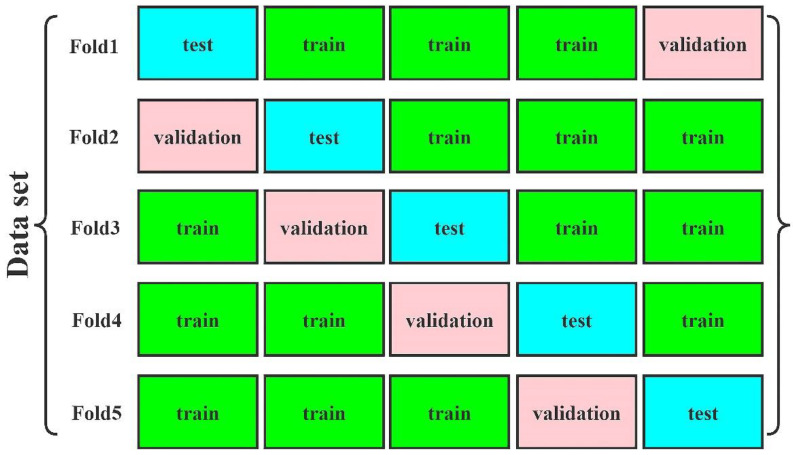
Schematic of the fivefold cross-validation. Train = train set, test = test set, validation = validation set.

**Figure 8 diagnostics-13-01067-f008:**
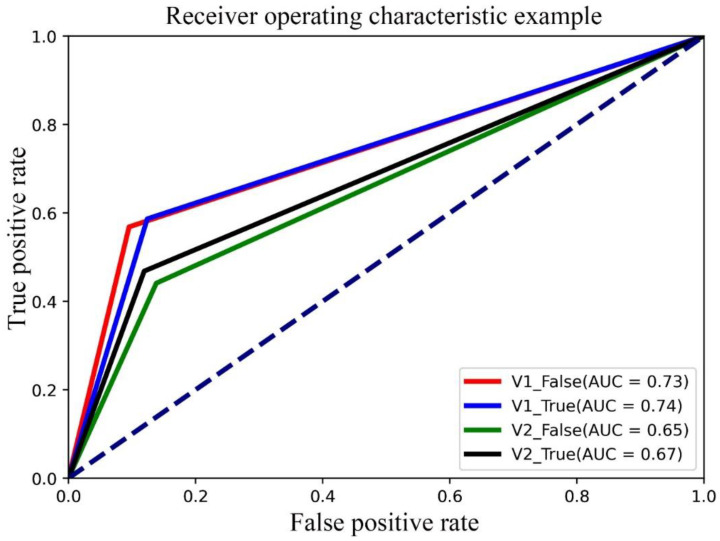
AUC analyses of the four proposed models (V1_False, V1_True, V2_False, V2_True) based on breast DCE-MRI. It was observed that the V1_True model (with AUC = 0.74) performed significantly better than the other three models on the validation set.

**Figure 9 diagnostics-13-01067-f009:**
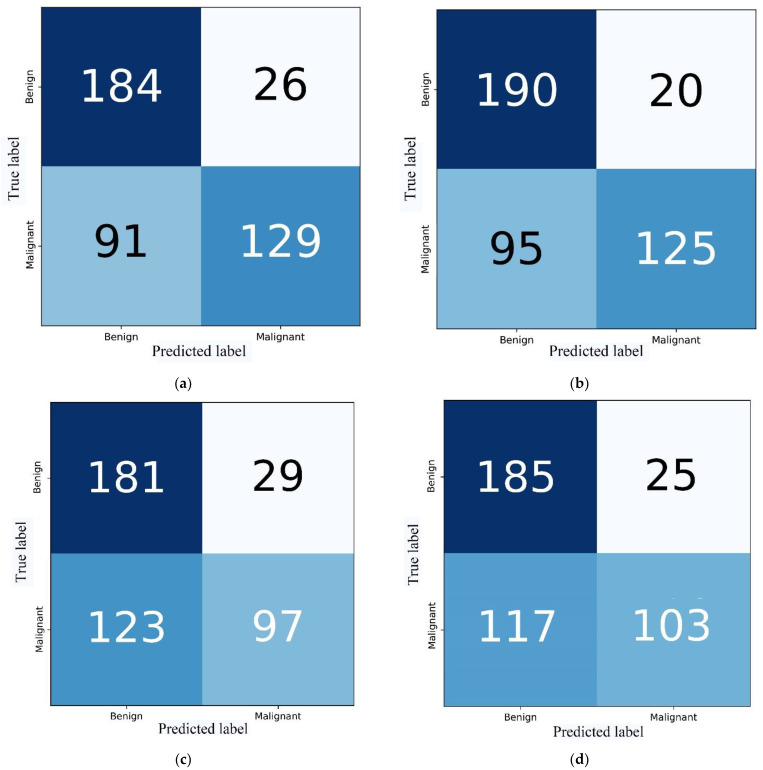
Confusion matrices of V1_False (**a**), V1_True (**b**), V2_False (**c**) and V2_True (**d**) on the validation set. Intuitively, we can see the different numbers of correctly versus erroneously predicted images for benign and malignant lesions in the validation set. B represents benign, and M represents malignant.

**Table 1 diagnostics-13-01067-t001:** Clinical information of the patients.

Pathological Diagnosis	Cases	Percent (%)	Age (Years)	Lesion Diameter (mm)
Malignant lesions			48.2 ± 11.4	24.00 ± 11.09
Invasive ductal carcinoma	124	80.52		
Intraductal carcinoma	19	12.34		
Invasive lobular carcinoma	4	2.60		
Mucinous carcinoma	4	2.60		
Lymphoma	1	0.65		
Papillary carcinoma	2	1.30		
Total	154	100.00		
Benign lesions			45.0 ± 10.5	32.89 ± 16.45
Cyst	17	9.83		
Adenosis	26	15.03		
Fibroadenoma	111	64.16		
Chronic inflammation	4	2.31		
Intraductal papilloma	13	7.51		
Lobular tumour	2	1.16		
Total	173	100.00		

**Table 2 diagnostics-13-01067-t002:** Dynamic contrast-enhanced magnetic resonance imaging parameters used in this study.

Parameters.	Philips Achieva	GE Healthcare
Field strength	3.0 T	3.0 T
No. of coil channels	8	8
Acquisition plane	Axial	Axial
Pulse sequence	3D gradient echo (Thrive)	Enhanced fast gradient echo 3D
Repetition time (ms)	5.5	9.6
Echo time (ms)	2.7	2.1
Flip angle	10°	10°
No. of postcontrast phases	5	5
Fat suppression	Yes	Yes
Scan time	570 s	500 s

Note. 3D, three-dimensional; ms, millisecond; s, second.

**Table 3 diagnostics-13-01067-t003:** Image augmentation parameters.

Parameter	Value
Rotation range	60°
Shear range	0.2
Zoom range	0.2
Horizontal flip	True
Vertical flip	True
Fill mode	Nearest

**Table 4 diagnostics-13-01067-t004:** Classification report for the DTL models on the validation set.

Models	Params1	Params2	Params3	Time (min)	Size (MB)
V1_False	3,228,864	0	3,228,864	19.23	19.4
V1_True	3,228,864	3,206,976	21,888	20.67	19.4
V2_False	2,257,984	0	2,257,984	27.55	16.7
V2_True	2,257,984	2,223,872	34,112	25.99	16.7

Note. params1, total number of parameters; params2, trainable parameters; params3, nontrainable parameters; time, the time taken to train the model; min, minute; size, the size of the file saved in h5 format.

**Table 5 diagnostics-13-01067-t005:** Results of the fivefold cross-validation.

Folds	Ac1	Loss1	Ac2	Loss2
Fold1	1.00	˂0.01	0.9815	0.2322
Fold2	1.00	˂0.01	0.9803	0.2225
Fold3	1.00	˂0.01	0.9812	0.2175
Fold4	1.00	˂0.01	0.9805	0.2402
Fold5	1.00	˂0.01	0.9814	0.2156

Note. Ac1, accuracy on the train set; loss1, loss on the train set; Ac2, accuracy on the test set; loss2, loss on the training test set.

**Table 6 diagnostics-13-01067-t006:** Classification report of the DTL models on the validation set.

DTL Models	Pr			Rc			F1			AUC
	Group1	Group2	Avg	Group1	Group2	Avg	Group1	Group2	Avg	
V1_False	0.88	0.59	0.77	0.67	0.83	0.73	0.76	0.69	0.73	0.73
V1_True	0.90	0.57	0.79	0.67	0.86	0.73	0.77	0.68	0.74	0.74
V2_False	0.86	0.44	0.74	0.60	0.77	0.65	0.70	0.56	0.66	0.65
V2_True	0.88	0.47	0.76	0.61	0.80	0.67	0.72	0.59	0.68	0.67

Note. group1, benign group; group2, malignant; avg, average.

## Data Availability

The datasets generated and/or analysed during the current study are available at https://github.com/HORIZ02/Mobilenet (accessed on 6 March 2023). The datasets are under continuous development and refinement.

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
