# Peer review of "Classification of Breast Lesions on DCE-MRI Data Using a Fine-Tuned MobileNet"

_diagnostics, 2023, doi:10.3390/diagnostics13061067_

Round 1

Reviewer 1 Report

Summary:

  • Long Wang and colleagues have presented the use of a MobileNet architecture to classify breast lesions based on DCE-MRI images. The use of mobile device hardware and the relevance of this article should not go understated. In total, 154 malignant lesions and 173 benign lesions were evaluated in this model, totaling 4350 images. The best-performing fine-tuned model was V1_True.
  • While I am in full support of the relevance and need for this manuscript’s publication, I have made several suggestions and points that need to be clarified below. The methods should be written with higher clarity and more detail to ensure replicability. Additionally, the author’s comment on concern regarding overfitting, yet despite a relatively good number of images (and overall patients) for a pilot study, had a very small (~10%) test and validation set.

Comments:

  • Few spelling and grammatical errors in the introduction, as well as reference inconsistencies. (Ex. Ref 19 should be at the end of the sentence); please revise
  • Breast MRI is referenced as DCE-BMRI or Breast DCE-MRI; would suggest one format is used
  • It is unclear what the P-value signifies in Table 1 and it lacks a standalone description of what the table represents or what this P-value means. 
  • I would ask that the authors clarify in the response and text their justification in the training and test sets. They claim the algorithm was overfitting, and given a test and validation set the size of 10% each, this is not surprising
  • The formatting of Figure 1 should be adjusted (letters on separate lines make this more difficult to read for the reader)
  • It is unclear from the authors whether segmentation / ROI was performed by a radiologist, or whether this was an unsupervised learning process and should be stated in the manuscript
  • 2.4 - “The images were randomly shuffled using a set of programs” - this should be corrected to highlight the method of randomization with a specific description of the randomization process or clarity on the program utilized
  • Other than rotation and flip, no other normalization was performed despite the use of two different machines with two different pulse sequences and repetition times
    • Can the authors please comment and clarify how normalization was pursued or why / why not normalization was performed?
  • If you are using standard evaluative metrics (ie. Ac, Rc, F1 score, FPR, and FNR) instead of placing all formulas in the main manuscript, I would suggest simple reference or inclusion in a supporting document separate from the main manuscript
  • How did you ensure that the difference in detection between malignant/benign was not simply just due to size comparison within the model (ie. first-order feature)?
  • Typo in Figure 6 (should be d-f, not d-e)

Author Response

Dear reviewer:

Reviewer 2 Report

I have carefully reviewed the article, I believe it is useful in the current discussion regarding the prevention of an important cancer problem worldwide. I therefore believe that it is of interest to readers and deserves publication. However I have some suggestions and requests:

- how is this application characterized by security profiles? with regard to clinical risk management are we sure that the system is reliable?

- what medico-legal problems could arise from its use? doi: 10.1186/s12913-018-3846-7.

- the bibliography should be expanded with other international studies

- the description of the methodology should be more detailed

Author Response

Dear reviewer:

Round 2

Reviewer 2 Report

the article is much improved after the suggestions. I think it can be published. I ask again attention to the editorial rules